# Exosome Release by Glucose Deprivation Is Important for the Viability of TSC-Null Cells

**DOI:** 10.3390/cells11182862

**Published:** 2022-09-14

**Authors:** Ji-Hyun Bae, Jong Hyun Kim

**Affiliations:** Department of Biochemistry, School of Medicine, Daegu Catholic University, Daegu 42472, Korea

**Keywords:** TSC, exosome, mTORC1, glucose deprivation, cell viability

## Abstract

The control of exosome release is associated with numerous physiological and pathological activities, and that release is often indicative of health, disease, and environmental nutrient stress. Tuberous sclerosis complex (TSC) regulates the cell viability via the negative regulation of the mammalian target of rapamycin complex (mTORC1) during glucose deprivation. However, the mechanism by which viability of TSC-null cells is regulated by mTORC1 inhibition under glucose deprivation remains unclear. Here, we demonstrated that exosome release regulates cell death induced by glucose deprivation in TSC-null cells. The mTORC1 inhibition by rapamycin significantly increased the exosome biogenesis, exosome secretion, and cell viability in TSC-null cells. In addition, the increase in cell viability by mTORC1 inhibition was attenuated by two different types of inhibitors of exosome release under glucose deprivation. Taken together, we suggest that exosome release inhibition might be a novel way for regression of cell growth in TSC-null cells showing lack of cell death by mTORC1 inhibition.

## 1. Introduction

In multicellular organisms, cells constantly send and receive particular signals to coordinate the actions of other organs, tissues, and cells. The ability to send special messages quickly and efficiently enables cells to coordinate their diverse functions, such as survival, proliferation, migration, differentiation, and even development [1,2,3,4]. Cell to cell communication in the tumor microenvironment is one of the important networks in order to support or halt tumor progression [5,6,7]. Interactions among neighboring cells in the primary tumor are essential for tumor growth and development, and intercellular communication also happens through a complex system involving secreted vesicles [8,9]. When a more complex message needs to be transferred to an adjacent or distant cell, exosomes, known as nano-size small membrane vesicles secreted from endosomal compartments called multivesicular bodies (MVBs), are employed [10,11]. Once released, the exosomes can both travel locally through the circulation, and reach distant sites. The characteristics of exosome-mediated communication are that messages can be delivered to multiple specific locations [12,13,14].

To date, a number of proteomics and transcriptomic profile analyses have been performed to understand the biological function of exosomes. Several researchers reported that the typical exosome maker proteins identified in exosome vesicles are tetraspanins such as CD9, CD63, and CD81 [15,16]. Other cytoplasmic proteins are annexin, Rab, Alix, and TSG101, which are cell-type specific marker proteins for the exosome-secreting cells [17,18]. As mentioned above, cells also convey diverse information to the microenvironment through exosomes and other extracellular vesicles (EVs) to intercommunicate through the complex signaling networks that are only beginning to be unveiled. Many studies involving several distinct cancer cells showed that tumor-derived exosomes can induce tumor cell proliferation [5,8,9]. For example, positive autocrine effects for proliferation were observed in chronic myeloid leukemia and human gastric cancer through signaling molecules and a long noncoding RNA [19]. In bladder cancer, glioblastoma, and prostate cancer, tumor-derived exosomes induced cell proliferation through triggering specific survival pathways [20,21,22]. The in vivo growth and tumor sizes of murine melanoma were increased, and melanoma tumor apoptosis was inhibited by systemic treatment of melanoma-derived exosome [23].

Tuberous sclerosis complex (TSC) such as TSC1 and TSC2, were initially identified as tumor suppressor genes in humans [24]. TSC1 and TSC2 attenuates cell proliferation and viability via the regulation of the mammalian target of rapamycin (mTORC1) as a master kinase of cell signaling [25]. This mTOR protein complex is stimulated by the coordination of several upstream signals, such as nutrients, growth factors, amino acids and glucose [26,27]. The level of glucose as a major energy source for biological life could determine the cell proliferation and survival via TSC-mTORC1 pathway [28]. TSC1 and TSC2 exist as a TSC1-TSC2 protein complex. Abnormal overactivation of the mTORC1 signaling pathway is caused by loss of function mutations in the TSC1 and TSC2 genes [29]. Therefore, targeting mTORC1 is one of the best ways to suppress the TSC function. The mTORC1 inhibitor, rapalogs, has shown surprising results in recent clinical trials, but tumors are alive again when the mTORC1 inhibitor is stopped [30,31]. During glucose deprivation, the viability of TSC-null cells was hardly affected by mTORC1 inhibition. Thus, identification of additional molecular connection and provision of effective therapeutic strategies are necessary for TSC patients [32,33].

In this study, we demonstrate that the exosome is implicated in the viability of TSC-null cells in glucose deprivation. We also suggest that therapeutic tools with exosome inhibitors are an effective way to target TSC-null cells that are hardly able to suppress cell proliferation by mTORC1 inhibition.

## 2. Materials and Methods

### 2.1. Reagents and Antibodies

Antibodies were purchased from the following companies: phosphor-S6K (Thr 389), S6K, mTORC1, pACC, ACC, caspase 3, cleaved PARP, TSC1, and TSC2 were purchased from Cell Signaling Technology, Danvers, MA, USA; beta-actin (ab8227) from Abcam, Cambridge, UK; beta-actin (937215) from R&D systems, Minneapolis, MN, USA; CD63 from Invitrogen, Waltham, MA, USA; CD81 from Santacruz; and goat anti-mouse IgG (H + L) secondary antibody (HRP), and goat anti-rabbit IgG (H + L) secondary antibody (HRP) from Thermo Fisher Scientific, Waltham, MA, USA. Reagents were obtained from the following sources: DAPI was obtained from Thermo Fisher Scientific; goat anti-mouse Alexa Fluor 488, goat anti-mouse Alexa Fluor 594, goat anti-rabbit Alexa Fluor 488, and goat anti-rabbit Alexa Fluor 594 were obtained from Invitrogen; phosphatase inhibitors, protease inhibitors, calpeptin, GW4869, and DMSO were obtained from Sigma-Aldrich, St. Louis, MO, USA; TSC1 and TSC2 siRNAs from Invitrogen; CCK-8 from Dojindo; rapamycin from Merck, Darmstadt, Germany; Lipofectamine 2000 and Exosome isolation buffer kit from Thermo Fisher Scientific; and glucose-free DMEM and high-glucose DMEM from Welgene, Gyeongsan, Korea.

### 2.2. Cell Culture

TSC1 (−/−) MEF cells and TSC2 (−/−) MEF cells were cultured in Dulbecco’s modified Eagle’s medium containing 10% fetal bovine serum (FBS, Welgene, Gyeongsan, Korea) and 1% penicillin-streptomycin at 37 °C in a 5% CO_2_. The TSC1 (−/−) MEF cells and TSC2 (−/−) MEF cells were kindly provided by Dr. David J. Kwiatkowski (Brigham and Women’s Hospital, Boston, MA, USA). The MEFs passage number was about 25–27 generations. No surface coating material was used in the study. Human embryonic kidney (HEK) 293 cells were cultured in Dulbecco’s modified Eagle’s medium containing 10% fetal bovine serum (FBS, Welgene, Gyeongsan, Korea) and 1% penicillin-streptomycin at 37 °C in a 5% CO_2_. The HEK293 cells were provided by Korea Cell Line Bank (KCLB, Seoul, Korea).

### 2.3. Chemical Treatments

For glucose starvation, TSC1 (−/−) MEF cells, TSC2 (−/−) MEFs cells, and HEK293 cells were twice rinsed with glucose-free DMEM. The cells were incubated with glucose-free DMEM containing DMSO or Rapamycin (100 nM) for 48 h. For Calpeptin (50 μM) or GW4869 (5 μM) treatment, TSC1 (−/−) MEF cells, TSC2 (−/−) MEF cells, and HEK293 cells were twice rinsed with glucose-free DMEM. The cells were incubated with glucose-free DMEM for the indicated times.

### 2.4. siRNA Treatment and Transfection

Specific siRNA duplexes corresponding to TSC1 and TSC2 were synthesized from Invitrogen (Waltham, MA, USA). The TSC1 siRNA sequence is 5′-CGACGUGACAGCUGUCUUUTT-3′ and TSC2 siRNA sequence is 5′-GCAUGGAAUGUGGCCUCAATT-3′. The stealth RNAi, a negative control duplex, is used as a scrambled control (Invitrogen). The scrambled siRNA did not match any sequence in the Blast Search Program. The siRNAs were diluted to a final concentration of 40 nM in serum-free DMEM (Welgene). These siRNAs were transfected into HEK293 cells using Lipofectamine 2000 (Invitrogen, Carlsbad, CA, USA), according to the manufacturer’s instructions. At 48 h after transfection, media were switched to complete DMEM for the indicated time.

### 2.5. Immunoblot Analysis

The TSC1 (−/−), TSC2 (−/−) MEFs cells, and HEK293 cells were harvested, washed twice with ice-cold PBS and lysed with lysis buffer (50 mM Tris/HCl, pH 7.5, 150 mM NaCl, 1% Triton X-100, 1% Na-deoxycholate, 0.1% SDS, 2 mM EDTA, inhibitor cocktail solution such as protease and phosphatase inhibitor cocktail). The cell lysate was centrifuged at 15,000× *g* for 15 min to obtain the supernatant. The supernatant was determined by Bradford assay to quantify protein amounts. The equal amounts of lysates were denatured by boiling at 95 °C for 5 min in a Laemmli sample buffer, separated by SDS 6% or 4–20% PAGE and finally transferred to nitrocellulose membranes. After blocking in PBST buffer (PBS + 0.05% Tween 20 + 5% skimmed milk powder), the membranes were incubated with the indicated individual monoclonal or polyclonal antibodies. The membranes were then reincubated with either anti-mouse or anti-rabbit IgG, coupled with horseradish peroxidase. Immunoreactivity was visualized by colorimetric reaction using an enhanced chemiluminescence substrate buffer (Thermo Scientific™, Waltham, MA, USA), according to the instructions of the manufacturer. The membranes were scanned by Davinch-Chemi Fluoro™ Imaging System (DaVinci-K, Geumchengu, Korea). Image quantification was expressed as mean ± SD by averaging three data. Images were quantified and measured using the UN-SCAN-IT gel program provided by DaVinci-K.

### 2.6. Immunocytochemistry

Approximately 5 × 10^4^TSC1 (−/−) MEF cells, or TSC2 (−/−) MEFs/well were seeded in a 24-well plate. After wiping a 15 × 15 mm cover glass with 70% ethanol, it was placed in a 24-well plate, and incubated with Poly-L-Lysine solution (Sigma-Aldrich, St. Louis, MO, USA) for at least 1 h. The primary antibody, CD63 (dilution 1:200) and beta-actin (dilution 1:500) were produced in rabbits, and CD81 (dilution 1:100) and beta-actin (dilution 1:500) were produced in mice. The sample was used by diluting the primary antibody in 1% BSA, and incubated overnight in a cold chamber at 4 °C. After washing 3 times with PBS, the secondary antibody was diluted and incubated for 1 h in the dark. Alexa594 labeled anti-rabbit and anti-mouse sera from goat (dilution 1:1000) and Alexa488 labeled anti-rabbit and anti-mouse sera from goat (dilution 1:1000). After washing the secondary antibody with PBS, DAPI (dilution 1:100) was diluted in PBS, and incubated for 1 h at RT. After dropping the mounting solution on the slide glass, covering with a coverslip, and sealing, the cells were checked with Nikon ECLIPSE Ti A1 confocal (Tokyo, Japan). Image quantification was expressed as mean + SD by selecting 4 or more cells, using ImageJ.

### 2.7. Exosome Purification and Isolation

Isolation of exosome was performed by modifying the exosome extraction method of Richard J. et al. [34]. All exosome isolations were performed from serum-free medium in TSC1 (−/−) MEF, TSC2 (−/−) MEF, and HEK293 cells. Serum-free culture media harvested from TSC1 (−/−) MEF, TSC2 (−/−) MEF, and HEK293 cells were centrifuged using a Hanil Union 32R (Gimpo, Korea) centrifuge at 300× *g* at 4 °C for 10 min to discard detached cells. The supernatant was harvested, and filtered through pore size 0.45 µM membrane filters (Merck Millipore) to remove contaminating apoptotic bodies and cell debris. The cleaned cell culture media were centrifuged at 100,000× *g* for 90 min at 4 °C with a Type 90 Ti rotor using a Beckman Coulter Optima^TM^ L-90K Ultracentrifuge. The exosome pellet was resuspended with 100 μL of lysis buffer.

### 2.8. Transmission Electron Microscopy (TEM)

The exosomes obtained through the exosome isolation process were resuspended in 200 µL of autoclaved distilled water. All carbon-coated copper grids were negatively stained with 10 µL of 2% uranyl acetate for 5 min at RT. The exosome was dropped on a 200-mesh copper grid, and dried at RT for 5 min. The grids were imaged by Hitachi 7700 transmission electron microscopy (Hitachi, Tokyo, Japan) at 50 kV. Images were captured with a side-mounted CCD camera (Gatan, Pleasanton, CA, USA).

### 2.9. Nanoparticle Tracking Analysis

Nanoparticle Tracking Analysis (NTA) from Malvern (NanoSight NS300, Worcestershire, UK) can measure the exosome size, concentration, and distribution in 10–1000 nm liquid, based on Brownian motion. After thoroughly washing the syringe pump with 1× PBS, 1 mL of exosomes diluted 10, 50, 100, or 500 times in PBS was added to the syringe to confirm, and an appropriate concentration was selected. The syringe was inserted into the NanoSight’s syringe pump. The exosomes were injected at a constant flow rate at RT. Brownian motions of particles (exosomes) present in the sample were subjected to a laser beam, recorded with a camera, and converted into size and concentration parameters by NTA via the Stokes–Einstein equation. Three videos were acquired for each sample.

### 2.10. Cell Viability Assay

Cell viability assay was performed according to the manufacturer’s protocol using Cell Counting Kit-8 (Dojindo, Rockville, MD, USA). A quantity of 2 × 10^4^ cells/well was seeded in the wells of 96-well plates at a volume of 200 µL/well and the chemicals were treated for 24 h. After 24 h, the media containing the drug were completely removed, after putting 200 µL of complete medium into each well, 10 µL of CCK-8 was added, and incubated at 37 °C for 2 h. The optical density (450 nm) representing cell viability was measured by Allsheng AMR-100 spectrophotometry (Hangzhou, China). The independent experiments were performed in triplicate.

### 2.11. Statistical Analysis

Quantification of immunoblot images was performed using UN-SCAN-IT gel program. Statistical analysis was performed using Microsoft Excel 2013 including Student’s *t*-test (two-tailed). All experiments were independently performed in at least 3 replications. Error bars in graphic data mean ± standard deviation (SD). Statistical significance was asserted when the *p* value was lower than 0.05. The exact *p* value was provided in the Figure legend.

## 3. Results

### 3.1. mTORC1 Inhibition Increases the Exosome Release of TSC2 (−/−) MEF Cells in Glucose Deprivation

The TSC1/2 protein complex plays a critical factor regulating the activity of the mTORC1 complex under various environmental conditions, such as limited growth factor or energy level [27]. Glucose is an important energy source to control cell proliferation and cell viability [28]. Recently, it has been reported that cell viability is connected with extracellular vesicles [35,36]. Thus, to investigate the effects of mTORC1 activity on exosome biogenesis by glucose deprivation in TSC-null cells, we first examined the levels of CD63 and CD81 known as exosome markers in TSC2 (−/−) MEF cells. CD63 and CD81 were increased by rapamycin treatment in the absence of glucose (Figure 1A). The pACC is a substrate of AMPK, which is known to be activated by glucose deprivation. The pACC was increased by glucose deprivation. The pS6K is a substrate of mTORC1, which is known to be inhibited by rapamycin. In our study, the pS6K was therefore inhibited by treatment of rapamycin. While CD63 had a 9.4-fold increase by rapamycin treatment in the absence of glucose when compared to that of DMSO treatment in glucose deprivation, CD81 had a 18.3-fold increase by rapamycin treatment in the absence of glucose when compared to that of DMSO treatment in glucose deprivation (Figure 1B). Other exosome markers such as Alix and TSG101 were increased under the same conditions (Appendix A). To examine the exosome release by rapamycin treatment in the absence of glucose, we isolated the exosome from culture media in TSC2 (−/−) cells. The CD63 and CD81 were detected in culture media treated with rapamycin, indicating that during glucose deprivation, exosome release was increased in the rapamycin treatment (Figure 1C). While CD63 showed 38.1-fold increase in culture media treated with rapamycin in the absence of glucose when compared to that of DMSO treatment in glucose deprivation, CD81 showed a 16.3-fold increase under the same conditions when compared to that of DMSO treatment in glucose deprivation (Figure 1D). Alix and TSG101 in culture media were highly detected in rapamycin treatment under glucose deprivation (Appendix A). We also investigated the effects of mTORC1 activity on exosome biogenesis under the same conditions using MEF-WT cells. The levels of CD63, CD81, Alix, and TSG101 were unaffected by rapamycin treatment in the absence of glucose (Appendix A). The levels of CD63, CD81, Alix and TSG101 in culture media were not changed under the same conditions (Appendix A). Next, we isolated extracellular vesicles (EV) using ultracentrifugation from TSC2 (−/−) cells to visualize EV particles. We isolated the EV in the presence of glucose, but failed to isolate EV particles in the absence of glucose, because few exosomes are released under these conditions. However, we succeeded in isolating EV in the absence of glucose with rapamycin, because it induced the increase of EV in the TSC2 (−/−) cells. There were few differences of size, or morphology of the isolated exosomes under these conditions (Figure 1E). Next, we analyzed the population size of EV isolated from TSC1 (−/−) MEF cells. Although we failed to detect the particle size of EV treated with DMSO in the absence of glucose, because there is little exosome release under these conditions, we successfully harvested the EV particles—treated either with DMSO in the presence of glucose, or with rapamycin in the absence of glucose—with a view to analyze them. There was little significant difference in the size and distribution of EV nanoparticles between those obtained from DMSO and those from rapamycin treatment (Figure 1F). To investigate the expression of CD63 and CD81 in subcellular locations, we performed immunocytochemistry. The fluorescence signals of CD63 and CD81 were enhanced by rapamycin treatment in the absence of glucose when compared to that of DMSO treatment in glucose deprivation (Figure 1G). The relative signal intensity of CD63 increased by around 2.2-fold by rapamycin treatment during glucose starvation when compared to that of DMSO treatment in glucose deprivation. The relative signal intensity of CD81 showed an approximately 2.6-fold increase in the same condition (Figure 1H). These results suggest that exosome release is enhanced by mTORC1 inhibition under the absence of glucose in TSC2 (−/−) MEF cells.

### 3.2. mTORC1 Inhibition Increases the Exosome Release of TSC1 (−/−) MEF Cells in Glucose Deprivation

Next, we investigated the release of exosome with TSC1 (−/−) MEF cells in the absence of glucose when attenuated by mTORC1 inhibitor, rapamycin. The expression levels of CD63 and CD81 in lysate of TSC1 (−/−) MEF cells were increased by rapamycin treatment in the absence of glucose (Figure 2A). Under these conditions, the phosphorylation levels of ACC and S6K were reflected in the treatment of glucose and rapamycin, respectively (Figure 2A). While CD63 showed a 31.9-fold increase by rapamycin treatment in the absence of glucose when compared to that of DMSO treatment in glucose deprivation, CD81 showed a 67.2-fold increase by rapamycin treatment in the absence of glucose when compared to that of glucose deprivation (Figure 2B). The levels of Alix and TSG101 were increased under the same conditions (Appendix A). To examine the exosome release by rapamycin treatment in the absence of glucose, we also isolated the exosome from culture media in TSC1 (−/−) cells. The CD63 and CD81 were detected in culture media treated with rapamycin, indicating that exosome release was increased in the rapamycin treatment during glucose deprivation (Figure 2C). While CD63 showed a 20.5-fold increase in culture media treated with rapamycin in the absence of glucose when compared to that of DMSO treatment in glucose deprivation, CD81 showed a 33.2-fold increase in the same condition when compared to that of DMSO treatment in glucose deprivation (Figure 2D). Alix and TSG101 in culture media were highly detected in rapamycin treatment under glucose deprivation conditions (Appendix A). We also isolated extracellular vesicles (EV) using ultracentrifugation from TSC1 (−/−) cells. We isolated the EV in the presence of glucose, but failed to isolate EV particles in the absence of glucose due to the few exosomes released under these conditions. However, we succeeded in isolating EV in the absence of glucose with rapamycin since it induced the increase of EV in the TSC1 (−/−) cells. There were few differences of size or morphology of isolated exosomes under these conditions, similarly to the results obtained from TSC2 (−/−) cells (Figure 2E). We also analyzed the population size of EV isolated from TSC2 (−/−) MEF cells. However, we failed to count the particle size of EV treated with DMSO in the absence of glucose, due to the lack of exosome release under these conditions. We eventually analyzed the EV particles treated with DMSO in the presence of glucose or rapamycin in the absence of glucose. There were little significant differences in the size or distribution of EV nanoparticles obtained from DMSO or rapamycin treatment (Figure 2F). To investigate the expression of CD63 and CD81 in subcellular compartments, immunocytochemistry was performed. The fluorescence signals of CD63 and CD81 were enhanced by rapamycin treatment in the asence of glucose when compared to that of DMSO treatment in glucose deprivation (Figure 2G). The relative signal intensity of CD63 showed an approximately 2.2-fold increase by rapamycin treatment during glucose starvation when compared to that of DMSO treatment in glucose deprivation. The relative signal intensity of CD81 showed an approximately 2.3-fold increase under the same conditions (Figure 2H). These results suggest that the exosome release is enhanced by mTORC1 inhibition under glucose deprivation in TSC1 (−/−) MEF cells.

### 3.3. Exosome Release Is Increased by mTORC1 Inhibition in Transient Silencing of TSC1/2

As mentioned earlier, when mTORC1 activity was inhibited in glucose deprivation, the exosome release was increased in TSC-null MEF cells, in which TSC activity is permanently abrogated (Figure 1 and Figure 2). Therefore, we next investigated the effects of exosome release on mTORC1 inhibition during glucose deprivation in the HEK293 cells transiently transfected with TSC1 or TSC2 siRNA. CD63 and CD81 were increased by rapamycin treatment in the absence of glucose when compared to that of DMSO treatment in glucose deprivation. The pACC was increased, and the pS6K was inhibited under the same conditions (Figure 3A). In the HEK293 cells transfected with TSC1 siRNA, CD63 and CD81 showed 154.5- and 485.7-fold increases, respectively, by rapamycin treatment in the absence of glucose when compared to that of DMSO treatment in glucose deprivation. In the HEK293 cells transfected with TSC2 siRNA, CD63 and CD81 showed 228.5- and 149.6-fold increases, respectively, by rapamycin treatment in the absence of glucose when compared to that of DMSO treatment in glucose deprivation (Figure 3B). Other exosome markers such as Alix and TSG101 were increased under the same conditions (Appendix A). Next, we also isolated the exosome from culture media in the HEK293 cells transiently transfected with TSC1 or TSC2 siRNA. The CD63 and CD81 were detected in culture media treated with rapamycin, indicating that exosome release was increased by the rapamycin treatment during glucose deprivation (Figure 3C). In TSC1 siRNA-transfected HEK293 cells, CD63 showed 71.5-fold and CD81 had 23.8-fold increases, respectively, in culture media treated with rapamycin in the absence of glucose when compared to that of DMSO treatment in glucose deprivation. In TSC2 siRNA-transfected HEK293 cells, CD63 showed a 48.5-fold while CD81 showed a 180.6-fold increase in culture media treated with rapamycin in the absence of glucose when compared to that of DMSO treatment in glucose deprivation (Figure 3D). On the other hand, the above experiments were performed under the same conditions using HEK293 cells transfected with scrambled siRNAs. The levels of Alix and TSG101 were unaffected under the same conditions (Appendix A). Alix and TSG101 in culture media were not changed in rapamycin treatment under glucose deprivation conditions (Appendix A). These results suggest that exosome release is also enhanced in the HEK293 cells transiently transfected with TSC siRNAs under the same conditions.

### 3.4. mTORC1 Inhibition Increases Cell Viability under Conditions of Glucose Deprivation 

Since mTORC1 inhibition induces the exosome release in TSC-null cells under conditions of glucose starvation, we investigated the relationship between cell viability and exosome release. We first examined the morphology prior to monitoring cell viability under conditions of glucose starvation. TSC2 (−/−) MEF cells lost cell volume, and also showed cell shrinkage and apoptotic cell bodies due to glucose starvation. However, the morphology of TSC2 (−/−) MEF cells seemed almost normal by rapamycin treatment in glucose starvation when compared to that of the presence of glucose (Figure 4A). In addition, the viability of TSC2 (−/−) MEF cells was increased by rapamycin treatment in glucose starvation when compared to that of DMSO treatment in glucose starvation (Figure 4B). Next, we utilized two different types of exosome release inhibitors to examine the effects of exosome on the cell viability. Calpeptin as a cell-permeable calpain inhibitor attenuates vesicle secretion or trafficking. GW4869 as a neutral sphingomyelinase inhibitor prevents the endosome biogenesis and intraluminal vesicle (ILV) formation. Calpeptin and GW4869 showed the morphology of cell shrinkage and apoptotic bodies in TSC2 (−/−) MEF cells (Figure 4A). The viability of TSC2 (−/−) MEF cells increased by rapamycin treatment in glucose starvation which was attenuated by two different types of exosome inhibitors, calpeptin or GW4869 (Figure 4B). We also investigated the cell morphology and cell viability in TSC1 (−/−) MEF cells. The morphology of TSC1 (−/−) MEF cells appeared almost normal by rapamycin treatment in glucose starvation when compared to that in the presence of glucose (Figure 4C). The viability of TSC1 (−/−) MEF cells was also increased by rapamycin treatment in glucose starvation when compared to that of DMSO treatment in glucose starvation (Figure 4D). Calpeptin and GW4869 showed the morphology of cell shrinkage and apoptotic bodies in TSC1 (−/−) MEF cells (Figure 4C). The viability of TSC1 (−/−) MEF cells increased by rapamycin treatment in glucose starvation which was attenuated by two different types of exosome inhibitors, calpeptin or GW4869 (Figure 4D). To compare the effect of cell viability on the transient TSC silencing with exosome inhibitors, we examined the cell viability in HEK293 cells transfected with TSC1 or TSC2 siRNAs during glucose starvation. The viability of HEK293 cells transfected with TSC1 or TSC2 siRNA was increased by rapamycin treatment in glucose starvation when compared to that of DMSO treatment in glucose starvation (Figure 4E). This increase of cell viability by rapamycin treatment in glucose starvation was also attenuated by two different types of exosome inhibitors, calpeptin or GW4869 (Figure 4E). To show the apoptosis pathway involved in the reduction of cell viability, we examined the apoptosis marker proteins. The cleaved caspase 3 and poly (ADP-ribose) polymerase (PARP) was decreased by rapamycin treatment in glucose starvation when compared to that of DMSO treatment in glucose starvation (Figure 4F). These apoptosis markers, cleaved caspase 3 and PARP, were also enhanced by two different types of exosome inhibitors, calpeptin or GW4869, in HEK293 cells transfected with TSC1 or TSC2 siRNA (Figure 4F). These results suggest that cell viability is increased by rapamycin treatment in glucose starvation and that this viability is controlled by exosome release in TSC-null cells.

## 4. Discussion

Under nutrient limiting conditions, mTORC1 is mainly inhibited through a TSC1/2-dependent mechanism. Thus, in TSC-null cells, mTORC1 activity is hyperactivated, despite nutrient starvation conditions [25,29]. mTORC1 inhibition can also prolong the survival of nutrient-stressed cells by inhibiting cell death via diverse mechanisms. On the other hand, exosome is associated with a number of physiological and pathological responses [37,38]. Exosome is known as a useful vesicle for communications among cells, and cell-to-cell interaction. Despite many years of study, our understanding of the basic biology of exosomes remains unclear. In this study, we demonstrate that exosome release is important for the TSC-null cell viability under glucose deprivation showing exosome inhibitor-induced cell death in energy-depleted condition.

Exosome release is regulated by mTORC1 activity in TSC-null cells under glucose deprivation. Rapamycin, the mTORC1 inhibitor, increased the levels of CD63, CD81, Alix, and TSG101 in exosome isolated from the media of TSC-null cells under glucose deprivation (Figure 1C,D and Figure 2C,D). Rapamycin also increased the levels of CD63, CD81, Alix, and TSG101 in cell lysates, as well as exosome release (Figure 1A,B,G,H and Figure 2A,B,G,H). Rapamycin is known as an inhibitor of protein translation via mTORC1 inhibition, so general protein synthesis is slightly decreased. However, surprisingly, exosome markers such as CD63, CD81, Alix, and TSG101 were increased even by treatment of rapamycin in glucose-deprivation condition. In addition, rapamycin prolonged the TSC-null cell viability under glucose deprivation as shown in the cell morphology, cell viability assay, and apoptosis marker proteins, such as cleaved caspase 3 and PARP (Figure 4). Two types of inhibitors of exosome release attenuated the cell viability induced by the treatment of rapamycin in glucose-starved TSC-null cells. These results suggest that the regulation of exosome release provide a therapeutic way of controlling TSC-null cell viability under conditions of glucose-deprivation.

We investigated the diverse effects and phenotypes with permanent TSC-null cells. To escape the bias of results performed with stable cell lines, we utilized the transient TSC1 or TSC2 knock-down cells. The results for exosome release or cell viability obtained from TSC-null cells were similar to those of the phenotypes obtained in HEK-293 cells transiently transfected with TSC1 or TSC2 siRNA (Figure 3).

There were few differences in the biological phenotypes between TSC1-null cells and TSC2-null cells. Biological phenotypes refer to the exosome size, exosome concentration according to size distribution, exosome image for transmission of electron microscopy (TEM), and exosome characteristics. However, contents in terms of individual macromolecules inside the exosome might be different between TSC1-null and TSC2-null cells according to the signal treatment. Further study is necessary to characterize the exosome contents (Figure 1 and Figure 2).

The TSC-null cell viability induced by rapamycin under conditions of glucose deprivation is involved in the caspase-3 or PARP cleavage pathway. The two different types of inhibitors of exosome release promoted the cell death, or apoptosis, under the same conditions, showing the increased level of active caspase 3 and cleaved PARP. Thus, these results imply that cell viability is important in cell-to cell communication via exosome release.

In summary, our study showed that exosome release is regulated by mTORC1 activity, and mTORC1 inhibition increased TSC-null cell viability in glucose deprivation. Thus, exosome release can regulate TSC-null cell viability and inhibitors of exosome release control the TSC-null cells. Evidence of lack of cell death by mTORC1 inhibition suggests that the control of exosome release might be a novel way for inducing regression of tumor growth in TSC-null cells.

## Figures and Tables

**Figure 1 cells-11-02862-f001:**
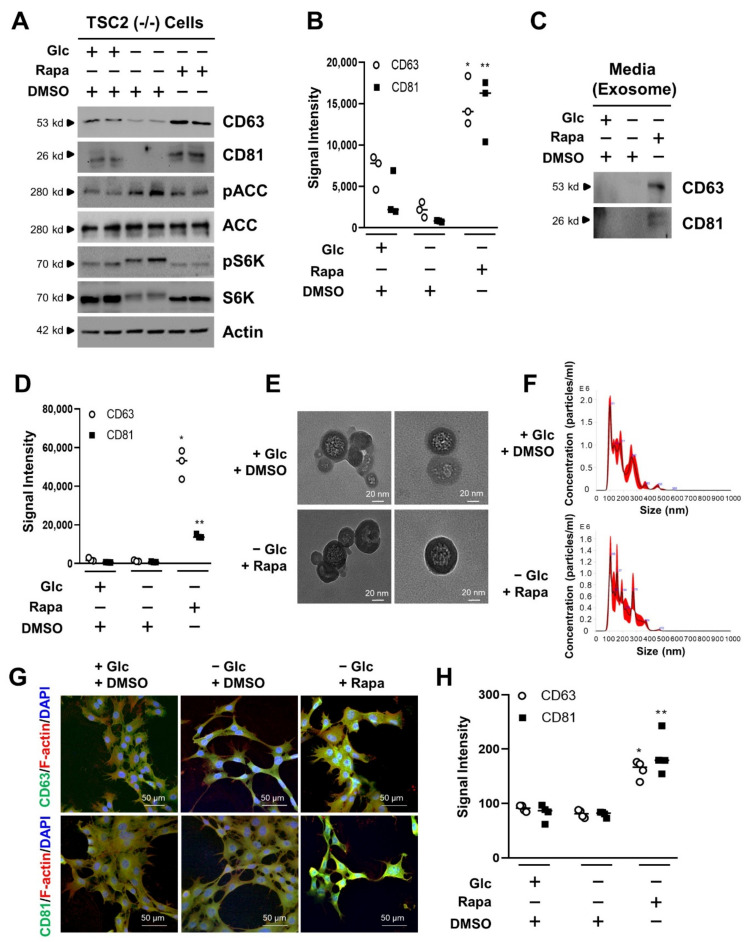
Effects of exosome release by mTORC1 inhibition in TSC2-null cells. (**A**) TSC2 (−/−) MEF cells were cultured with completed media. During serum starvation, cells were cultured in the absence or presence of glucose for 48 h. Rapamycin (100 nM) was treated in the absence of glucose for 24 h. The cells were harvested, lysed, and analyzed by immunoblotting with the indicated antibodies. Actin was used as a loading control. (**B**) The dot plot graph was quantified from three-independent immunoblotting data. The CD63, and CD81 are exosome marker proteins. These proteins were quantified to improve accuracy and show the measurable value. The quantification was performed with UN-SCAN-IT gel program. The detailed method is described in the materials and methods. There are significant differences between −Glc and −Glc + Rapa treatment groups. ** p* = 0.00203 for CD63 and *** p* = 0.00320 for CD81. (**C**) TSC2 (−/−) MEF cells were cultured with completed media. During serum starvation, cells were cultured in the absence or presence of glucose for 48 h. Rapamycin (100 nM) was treated in the absence of glucose for 24 h. After the collection of medium from cells, the media were subjected to ultracentrifugation to harvest the exosome fraction, as according to the materials and methods. The samples were analyzed by immunoblotting with antibodies for exosome marker proteins, such as CD63, and CD81. (**D**) The dot plot graph was quantified from three-independent immunoblotting data to improve the accuracy. The quantification was performed with UN-SCAN-IT gel program. The detailed method is described in the materials and methods. There are significant differences between −Glc and −Glc + Rapa treatment groups. ** p* = 0.000301 for CD63 and *** p* = 0.00002 for CD81. (**E**) Transmission electron microscopy (TEM) of extracellular vesicle for TSC2 (−/−) MEF cells. The images show the presence of glucose and the absence of glucose with rapamycin for extracellular vesicle. The scale bar is 20 nm. (**F**) The nanoparticle size of extracellular vesicle isolated from TSC2 (−/−) MEF cells was measured by NanoSight NS3000 from Malvern. The black graph indicates an average finite track length adjustment (FTLA) concentration/size. Red color shows the standard errors of mean. (**G**) Immunofluorescence images of TSC2 (−/−) MEF cells. The cells were fixed and stained with either anti-CD63 (green), anti-F-actin (red), and DAPI for nuclei (blue), on the one hand; or with anti-CD81 (green), anti-F-actin (red), and DAPI (blue), on the other. Stained cells were examined by confocal microscopy. The scale bar is 50 µm. (**H**) Quantitative presentations of the number of CD63, or CD81 in rapamycin- or DMSO-treated cells. A total of 20 randomly selected cells from three independent experiments were analyzed. There are significant differences between −Glc and −Glc + Rapa treatment groups. ** p* = 0.00041 for CD63 and *** p* = 0.00003 for CD81. Glc: Glucose, Rapa: Rapamycin, DMSO: Dimethyl Sulfoxide.

**Figure 2 cells-11-02862-f002:**
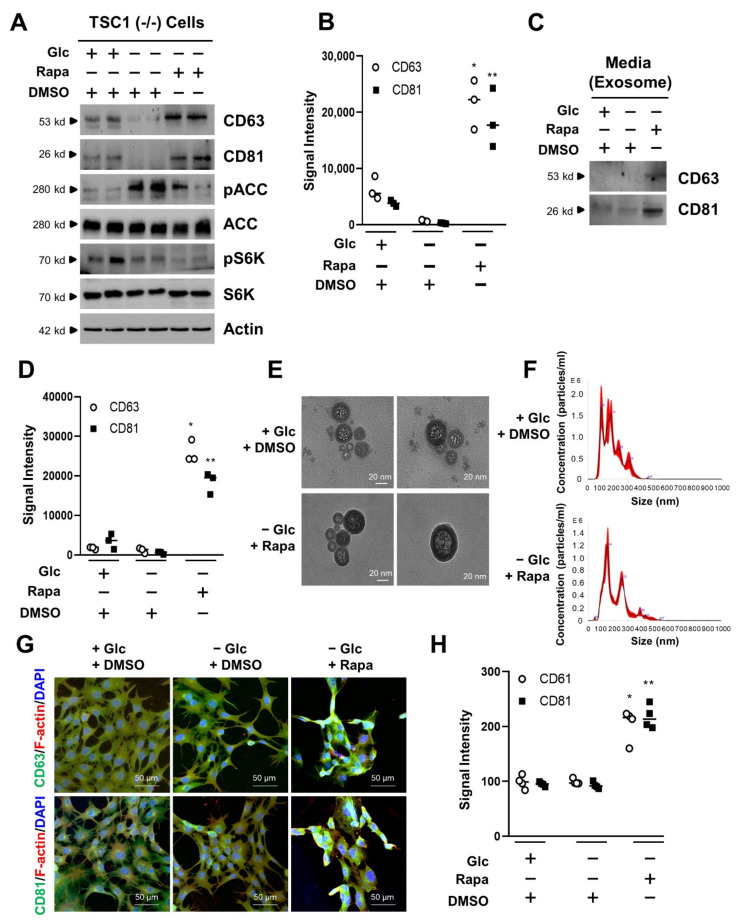
Effects of exosome release by mTORC1 inhibition in TSC1-null cells. (**A**) TSC1 (−/−) MEF cells were cultured with completed media. During serum starvation, cells were cultured in the absence or presence of glucose for 48 h. Rapamycin (100 nM) was treated in the absence of glucose for 24 h. The cells were harvested, lysed, and analyzed by immunoblotting with the indicated antibodies. Actin was used as a loading control. (**B**) The dot plot graph was quantified from three-independent immunoblotting data. The CD63 and CD81 are exosome marker proteins. The quantification was performed with UN-SCAN-IT gel program. There are significant differences between −Glc and −Glc + Rapa treatment groups. ** p* = 0.00117 for CD63 and *** p* = 0.00369 for CD81. (**C**) TSC1 (−/−) MEF cells were cultured with completed media. During serum starvation, cells were cultured in the absence or presence of glucose for 48 h. Rapamycin (100 nM) was treated in the absence of glucose for 24 h. After the collection of medium from cells, the media were subjected to ultracentrifugation to harvest the exosome fraction, according to the materials and methods. The samples were analyzed by immunoblotting with antibodies for exosome marker proteins, such as CD63, and CD81. (**D**) The dot plot graph was quantified from three-independent immunoblotting data. The quantification was performed with UN-SCAN-IT gel program. There are significant differences between −Glc and −Glc + Rapa treatment groups. ** p* = 0.00012 for CD63 and *** p* = 0.00034 for CD81. (**E**) Transmission electron microscopy (TEM) of the extracellular vesicle for TSC1 (−/−) MEF cells. The images show the presence of glucose and the absence of glucose with rapamycin for extracellular vesicles. The scale bar is 20 nm. (**F**) The nanoparticle size of extracellular vesicle isolated from TSC1 (−/−) MEF cells was measured with NanoSight NS3000 from Malvern. The black graph indicates an average finite track length adjustment (FTLA) concentration/size. Red color shows the standard errors of mean. (**G**) Immunofluorescence images of TSC1 (−/−) MEF cells. The cells were fixed and stained with either: anti-CD63 (green), anti-F-actin (red), and DAPI for nuclei (blue), on the one hand; or anti-CD81 (green), anti-F-actin (red), and DAPI (blue), on the other. Stained cells were examined by confocal microscopy. The scale bar is 50 μm. (**H**) Quantitative presentations of the number of CD63, or CD81 in rapamycin- or DMSO-treated cells. A total of 20 randomly selected cells from three independent experiments were analyzed. There are significant differences between −Glc and −Glc + Rapa treatment groups. ** p* = 0.00010 for CD63 and *** p* = 0.00125 for CD81. Glc: Glucose, Rapa: Rapamycin, DMSO: Dimethyl Sulfoxide.

**Figure 3 cells-11-02862-f003:**
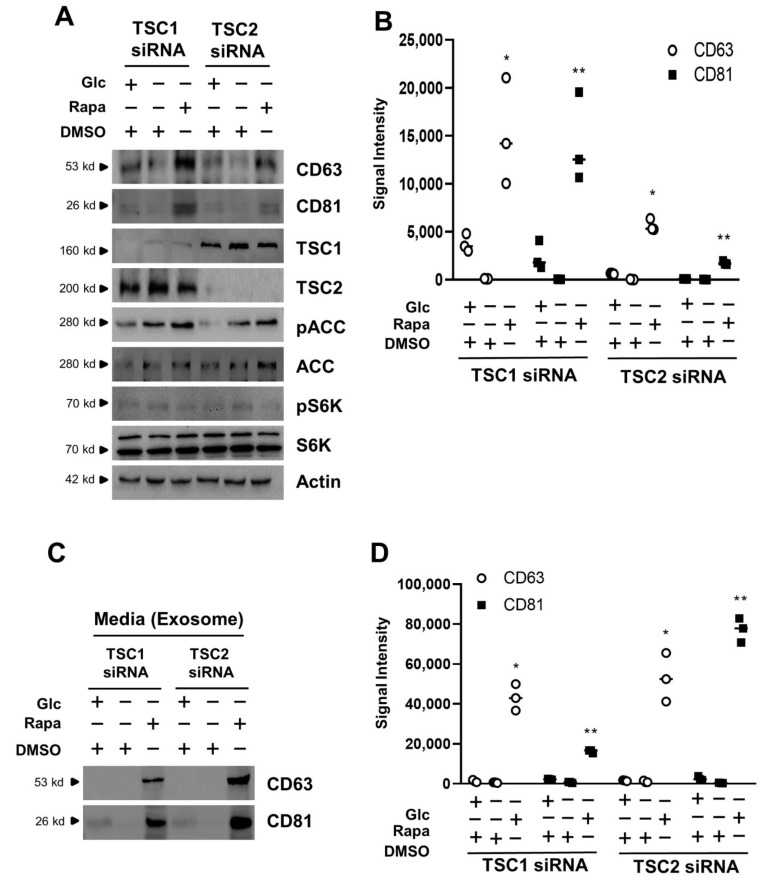
Exosome release is enhanced by mTORC1 inhibition in TSC1- or TSC2-silencing cells. (**A**) HEK293 cells transfected with either TSC1 siRNA or TSC2 siRNA were cultured with completed media, and then incubated in the absence or presence of glucose for 48 h during serum starvation. The cells incubated in the absence of glucose were treated with 100 nM rapamycin for 24 h. The cells were harvested and lysed, and analyzed by immunoblotting with the indicated antibodies. pACC was used as a marker for glucose deprivation. pS6K was used as an indicator of mTORC1 activity for rapamycin treatment. Actin was used as a loading control. (**B**) The dot plot graph was quantified from three independent immunoblotting data. The CD63, and CD81 are exosome marker proteins. The quantification was performed with UN-SCAN-IT gel program. There are significant differences between −Glc and −Glc + Rapa treatment groups. ** p* = 0.00952 for CD63 and *** p* = 0.00625 for CD81 in TSC1 siRNA, ** p* = 0.00010 for CD63 and *** p* = 0.00007 for CD81 in TSC2 siRNA. (**C**) HEK293 cells transfected with either TSC1 siRNA or TSC2 siRNA were cultured with completed media, and then incubated in the absence or presence of glucose for 48 h during serum starvation. After the collection of medium from cells, the media were subjected to ultracentrifugation to harvest the exosome fraction, according to the materials and methods. The samples were analyzed by immunoblotting with antibodies for exosome marker proteins, such as CD63, and CD81. (**D**) The dot plot graph was quantified from three independent immunoblotting data. Quantification of the data of CD63 and CD81 was performed with UN-SCAN-IT gel program. There are significant differences between −Glc and −Glc + Rapa treatment groups. ** p* = 0.00001 for CD63 and *** p* = 0.00625 for CD81 in TSC1 siRNA, ** p* = 0.00179 for CD63 and *** p* = 0.00003 for CD81 in TSC2 siRNA. Glc: Glucose, Rapa: Rapamycin, DMSO: Dimethyl Sulfoxide.

**Figure 4 cells-11-02862-f004:**
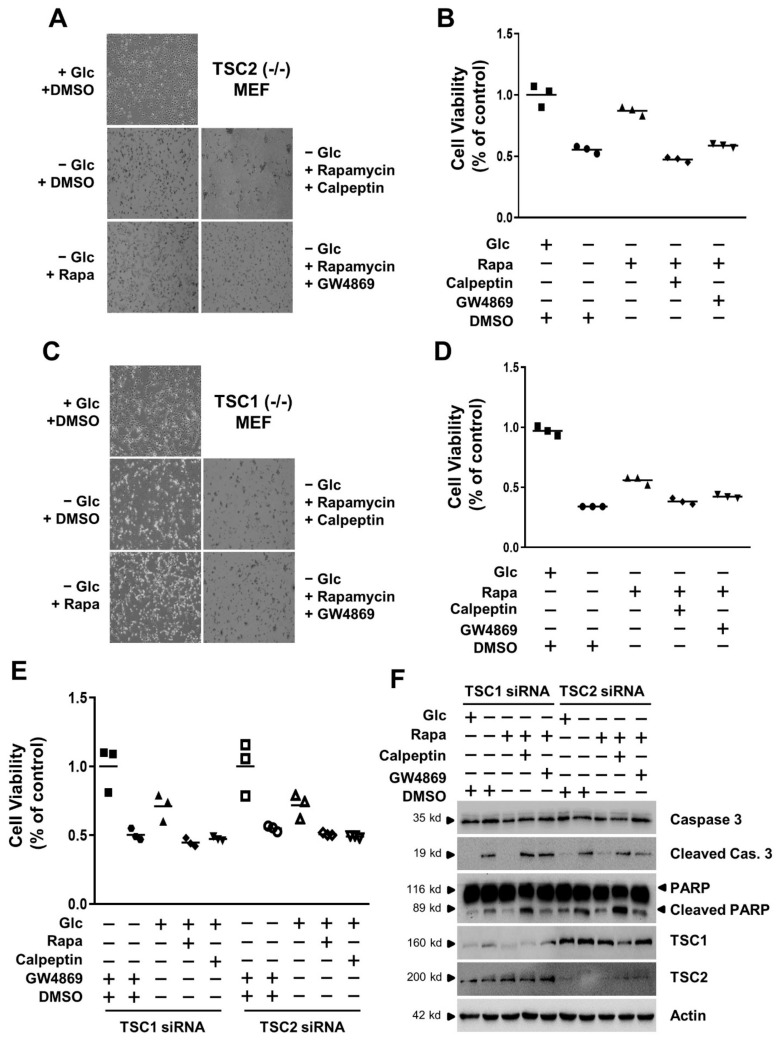
The mTORC1 inhibition-induced cell viability is suppressed by exosome release inhibitors in glucose deprivation. (**A**) During serum starvation, TSC2 (−/−) MEF cells were cultured in the absence or presence of glucose for 48 h. The cells incubated in the absence of glucose were treated with the combination of 100 nM rapamycin, and either 50 μM calpeptin or 5 μM GW4869 for 24 h. These representative images were taken from random sites after treatment. (**B**) TSC2 (−/−) MEF cells were cultured in the absence or presence of glucose for 48 h. The cells incubated in the absence of glucose were treated with the combination of 100 nM rapamycin, and either 50 μM calpeptin or 5 μM GW4869 for 24 h. Then, the cell viability was measured with CCK8 reagents. The detailed method is described in the materials and methods. (**C**) During serum starvation, TSC1 (−/−) MEF cells were cultured in the absence or presence of glucose for 48 h. The cells incubated in the absence of glucose were treated with the combination of 100 nM rapamycin, and either 50 μM calpeptin or 5 μM GW4869 for 24 h. These representative images were taken from random sites after treatment. (**D**) TSC1 (−/−) MEF cells were cultured in the absence or presence of glucose for 48 h. The cells incubated in the absence of glucose were treated with the combination of 100 nM rapamycin, and either 50 μM calpeptin or 5 μM GW4869 for 24 h. Then, the cell viability was measured with CCK8 reagents. (**E**) HEK293 cells transfected with either TSC1 siRNA or TSC2 siRNA were cultured with completed media and then incubated in the absence or presence of glucose for 48 h during serum starvation. The cells incubated in the absence of glucose were treated with the combination of 100 nM rapamycin, and either 50 μM Calpeptin or 5 μM GW4869 for 24 h. Then, the cell viability was measured with CCK8 reagents. (**F**) The HEK293 cell obtained with the same methods in panel E were harvested, lysed, and analyzed by immunoblotting with the indicated antibodies. Actin was used as a loading control. Glc: Glucose, Rapa: Rapamycin, DMSO: Dimethyl Sulfoxide.

## Data Availability

The data that support the findings of this study are available upon reasonable request from the authors.

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
