# Peer review of "Exosome Release by Glucose Deprivation Is Important for the Viability of TSC-Null Cells"

_cells, 2022, doi:10.3390/cells11182862_

Round 1
Reviewer 1 Report
page 2 lines 68 to 70
“…we demonstrate that the exosome is implicated” do you mean the exosome population or all exosomes.
“…with exosomes inhibitors is an effective way to TSC-null cells..” I think you are missing a word or words. Did you mean to add “exosome inhibitors are an effective way to target TSC-null cells”?
Concerns:
The TSC1-/- and TSC2-/- MEFs experiments are not compared to WT MEFs. How do these conditions affects WT MEFs? How is exosome release affected in normal MEFs by these conditions?
HEK293 siTSC1 and siTSC2 experiments are not compared to siScramble cells. How do these conditions affect siScramble cells? How do these conditions affect exosome release in these cells?
Why not use TSC null cancer cell lines? How would TSC mutations affect exosome release and would these conditions to suppress exosome release also target cancer cells with TSC mutants or only TSC-null cells?
Author Response
Dear Dr. Editor
Thank you for giving us the opportunity to revise our manuscript entitled “Exosome Release by Glucose Deprivation Is Important for the Cell Viability of TSC-null Cells” (Manuscript Number: Cells-1853422). We appreciate the reviewer’s constructive and insightful comments. We have revised the manuscript accordingly, and would like to re-submit it for your consideration. We have addressed the comments raised by the reviewers, and the amendments are highlighted in red in the revised manuscript. We hope that the revised version of the manuscript is now acceptable for publication in Cells journal. Point-by-point responses to the reviewers’ comments are listed below this letter.
Reviewer 1
page 2 lines 68 to 70
“…we demonstrate that the exosome is implicated” do you mean the exosome population or all exosomes.
Answer: We believe that exosome means a subpopulation of extracellular vesicles (EV). We don’t think that all exosome play the critical roles in viability of TSC-null cells. Considering that certain factrors in some exosomes in glucose deprivation conditions may modulate cell viability, we would like to identify the specific factors to start our next projects.
“…with exosomes inhibitors is an effective way to TSC-null cells..” I think you are missing a word or words. Did you mean to add “exosome inhibitors are an effective way to target TSC-null cells”?
Answer: Thank you for your kind comment. We added “to target TSC-null cells but not TSC-null cells”.
Concerns:
- The TSC1-/- and TSC2-/- MEFs experiments are not compared to WT MEFs. How do these conditions affects WT MEFs? How is exosome release affected in normal MEFs by these conditions?
Answer: Thank you for your kind comment. We performed the corresponding experiment to the reviewer’s request using MEF-WT cells under the same condition. The levels of CD63 and CD81 used in original version and the levels of Alix and TSG101 added in revised version were unaffected under the same condition in MEF-WT cells when compared to that of TSC2-/- and TSC1-/- cells. Exosome release was also unaffected under the same condition in MEF-WT cells. We described the above in result section and added the explanation in Supple. Figure 2. (Please see the paragraph marked in red on page 5, line 12-16)
- HEK293 siTSC1 and siTSC2 experiments are not compared to siScramble cells. How do these conditions affect siScramble cells? How do these conditions affect exosome release in these cells?
Answer: Thank you for your kind comment. We performed the corresponding experiment using HEK293 cells transfected with scrambled siRNAs under the same conditions. The levels of CD63 and CD81 as well as levels of Alix and TSG101 were unaffected under the same condition in HEK293 cells transfected with scrambled siRNAs when compared to that of HEK293 transfected cells with either TSC2 or TSC1 siRNAs. Exosome release was not changed in MEF-WT cells under the same condition. We described the above in result section and added the explanation in Supple. Figure 5. (Please see the paragraph marked in red on page 9, line 7 from bottom)
- Why not use TSC null cancer cell lines? How would TSC mutations affect exosome release and would these conditions to suppress exosome release also target cancer cells with TSC mutants or only TSC-null cells?
Answer: We really appreciate you. As you know, TSC composed of a heterodimer of TSC1 and TSC2, acts as a negative regulator of mTOR signaling. In this study, we discovered the role of exosome by glucose deprivation in TSC-null cells. We believe that TSC mutation can induce the loss of function in TSC that mimics the phenotypes of TSC-deficient cells. Thus, if we utilize the stable cell lines derived from TSC mutations which induce the function of TSC-null, we speculate the similar result obtained from TSC-null cells including exosome release, TEM images, cell morphology, and cell viability. However, stable cell lines of this kinds have not been available to us. So, we are still not sure and have any evidence to support our suggestion yet.

Reviewer 2 Report
In this article, the authors claim that the release of exsosomes *(which are a subpopulation of extracellular vesicles; EVs) is increased under glucose deprivation and is linked with cell viability of TSC-null cells.
Major concern:
Although the topic is interesting, there is fundamental flaw in the experimental design where a critical control is missing throughout the entire manuscript. While authors examined the EV release by mTORC1 inhibition under the absence/deprivation of glucose and observed that EV release was increased. It is mandatory to validate the hypothesis by mTORC1 inhibition under normal glucose conditions also.
Specific comments:
1. Results section 3.1 and 3.2: mTORC1 inhibition increases the exosome release of TSC2 (-/-) and TSC1 (-/-) MEF cells in glucose deprivation and the related results which claim that the levels of EVs and EV markers increased by rapamycin treatment in the absence/deprivation of glucose.
An important control is missing here. Since several studies have shown that EV release is increased or inhibited under stress conditions, and authors of current study have applied two stress conditions (mTORC1 inhibition, and glucose deprivation) it is imperative to validate whether the increase in EV release is due to mTORC1 inhibition inhibition itself or rather by glucose. Thus, authors should perform mTORC1 inhibition under normal glucose concentration and under glucose deprivation separately, and measure the levels of EVs and EV markers in both conditions. without performing this control experiment it is cannot be claimed what was the factor for the increase of EV release.
2. Similarly, experiments related to mTORC1 inhibition and TSC1/2 silencing should be performed mTORC1 inhibition with TSC1/2 silencing and without silencing TSC1/2. Without counter controls it not possible to judge the conclusion.
3. The linked cell viability assays and downstream pathways should be validated under both conditions.
4. Cell viability by mTORC1 inhibition was attenuated by two different types of inhibitors of exosome release under glucose deprivation. How was the cell viability by mTORC1 inhibition by two different type inhibitors of exosome release under normal glucose?. This is an important counter control experiment to validate the hypothesis.
5. In the results section at page 4, the authors have mentioned that reference 2-8 (EVs) are relate to cell viability. These references do not report the cell viability is affected by EVs. Authors should cite original articles which have reported the roles of EVs in cell viability.
6. Statistical differences are not annotated on figures/bar charts, and also the name of statistical parameter used in each case is not mentioned in the legends.
7. Instead of general bar graphs, the data should be presented by dot plots to show the position/distribution of individual biological replicates.
8. The whole electrophoretogram/ blot images should be provided in the supplementary material to judge the results.
9. On a separate note: the EV investigations should follow MISEV2018 guidelines published by International Society for Extracellular Vesicles (ISEV), (PMID: 30637094). It is not justified why authors call their isolated fractions exosomes but not the extracellular vesicles (EVs)?. To date there is not a single method available which could isolate exosomes alone. The isolated fraction is always a heterogenous population of EVs, no matter which method was used, and which material was used. Also, the markers such as CD63, CD9, CD81 and HSP70 etc are common for both exosomes and ectosomes (microvesicles). Similarly, the diameter or size detected by size detection instruments, cannot distinguish exosomes from microvesicles of same size. some microvesicles also show smaller diameters, like exosomes (thus EVs including small microvesicles and exosomes are called small EVs). If authors want to claim that the isolated fractions are exosomes then according to MISEV2018 guidelines (PMID: 30637094), at least two endosomal markers e.g. ALIX and TSG101 etc., should be identified along with surface markers (CD63, CD9,)., in order to verify that the vesicles are of endosomal origin i.e., exosomes. Otherwise, apply the term extracellular vesicles instead of exosomes.
Others:
Although not of major concern, but import for readers, there are enormous errors in the text. Few examples are provided below but are not limited to;
i. lack of nutrient, glucose, tuberous sclerosis complex (TSC) regulates – revise the phrase. Should it be…. tuberous sclerosis complex (TSC) is regulated by lack of nutrient, such as glucose via negative regulation of………….?
ii. two different type inhibitors – two different types of inhibitors
iii. page 2, line 61: targeting mTORC1 becomes a the most available….
iv. method of Richard J → Richard J et al.,
Author Response
Dear Dr. Editor
Thank you for giving us the opportunity to revise our manuscript entitled “Exosome Release by Glucose Deprivation Is Important for the Cell Viability of TSC-null Cells” (Manuscript Number: Cells-1853422). We appreciate the reviewer’s constructive and insightful comments. We have revised the manuscript accordingly, and would like to re-submit it for your consideration. We have addressed the comments raised by the reviewers, and the amendments are highlighted in red in the revised manuscript. We hope that the revised version of the manuscript is now acceptable for publication in Cells journal. Point-by-point responses to the reviewers’ comments are listed below this letter.
Reviewer 2
In this article, the authors claim that the release of exsosomes *(which are a subpopulation of extracellular vesicles; EVs) is increased under glucose deprivation and is linked with cell viability of TSC-null cells.
Major concern:
Although the topic is interesting, there is fundamental flaw in the experimental design where a critical control is missing throughout the entire manuscript. While authors examined the EV release by mTORC1 inhibition under the absence/deprivation of glucose and observed that EV release was increased. It is mandatory to validate the hypothesis by mTORC1 inhibition under normal glucose conditions also.
Specific comments:
- Results section 3.1 and 3.2: mTORC1 inhibition increases the exosome release of TSC2 (-/-) and TSC1 (-/-) MEF cells in glucose deprivation and the related resultswhich claim that the levels of EVs and EV markers increased by rapamycin treatment in the absence/deprivation of glucose.
An important control is missing here. Since several studies have shown that EV release is increased or inhibited under stress conditions, and authors of current study have applied two stress conditions (mTORC1 inhibition, and glucose deprivation) it is imperative to validate whether the increase in EV release is due to mTORC1 inhibition inhibition itself or rather by glucose. Thus, authors should perform mTORC1 inhibition under normal glucose concentration and under glucose deprivation separately, and measure the levels of EVs and EV markers in both conditions. without performing this control experiment it is cannot be claimed what was the factor for the increase of EV release.
Answer: Thank you for your comment. However, we do not completely agree with the reviewer’s claims. We do not claim that mTOR inhibition under normal glucose conditions reduces cell apoptosis via exosome release, but rather that mTOR inhibition reduces cell apoptosis in glucose deprivation conditions. We compared the normal or inhibitory activity of mTOR under glucose deprivation in TSC-null cells. In fact, we did not observe the significant changes of exosome marker proteins such as CD63, CD81, Alix, and TSG101, and exosome release by mTOR inhibition under normal glucoses condition in TSC-WT cells (see, figure below). Thus, we, first, focused on the exosome release by mTOR inhibition under glucose deprivation and then discovered that mTOR inhibition during glucose deprivation modulates the cell viability through exosome release. As suggested in the title, we are claiming that mTOR inhibition in the absence of glucose increases exosome secretion to decrease cell apoptosis, thereby increasing cell survival. Therefore, we believe the reviewer’s claims is out of scope of which we’re addressing main issues.
- Similarly, experiments related to mTORC1 inhibition and TSC1/2 silencing should be performed mTORC1 inhibition with TSC1/2 silencing and without silencing TSC1/2. Without counter controls it not possible to judge the conclusion.
Answer: Answer: Thank you for your kind comment. We performed the corresponding experiment to the reviewer’s request using MEF-WT cells under the same condition. The levels of CD63 and CD81 used in original version and the levels of Alix and TSG101 added in revised version were unaffected under the same condition in MEF-WT cells when compared to that of TSC2-/- and TSC1-/- cells. Exosome release was also unaffected under the same condition in MEF-WT cells. We described the above in result section and added the explanation in Supple. Figure 2. (Please see the paragraph marked in red on page 5, line 12-16)
- The linked cell viability assays and downstream pathways should be validated under both conditions.
Answer: As described above, we focused on the exosome release or cell viability under the glucose deprivation conditions. We do not insist that our hypothesis is the role of exosome on cell viability under normal glucose conditions. Thus, we believe the reviewer’s comment is outside of the scope of our proposal. In addition, we show the activation of caspase 3 and PARP as the downstream pathway in glucose deprivation conditions (Figure 4F in revised version).
- Cell viability by mTORC1 inhibition was attenuated by two different types of inhibitors of exosome release under glucose deprivation. How was the cell viability by mTORC1 inhibition by two different type inhibitors of exosome release under normal glucose?. This is an important counter control experiment to validate the hypothesis.
Answer: As described above, we focused on the role of exosome on cell viability under the glucose deprivation conditions. In this condition, we utilized the mTOR inhibitors and two different types of inhibitors of exosome release because we want to clarify the roles of exosome, and mTOR on cell viability under glucose deprivation conditions. We do not insist that our hypothesis is the cell viability related with exosome under normal glucose conditions. Thus, we believe the reviewer’s comment is out of the scope of our proposal.
- In the results section at page 4, the authors have mentioned that reference 2-8 (EVs) are relate to cell viability. These references do not report the cell viability is affected by EVs. Authors should cite original articles which have reported the roles of EVs in cell viability.
Answer: Thank you for your comment. We referred that the cell viability is related with the extracellular vesicles. We inserted the following references in the revised version.
- Hewson C,; Capraro D,; Burdach J,; Whitaker N,; Morris KV. Extracellular vesicle associated long non-coding RNAs functionally enhance cell viability. Noncoding RNA Res. 2016, 1, 3-11.
- Lee SS,; Won JH,; Lim GJ,; Han J,; Lee JY,; Cho KO,; Bae YK. A novel population of extracellular vesicles smaller than exosomes promotes cell proliferation. Cell Commun Signal. 2019, 17, 95.
- Statistical differences are not annotated on figures/bar charts, and also the name of statistical parameter used in each case is not mentioned in the legends.
Answer: Thank you for your kind comment. We replaced the bar graphs into dot plots and showed p values and statistical analysis used in “figure legend” and “Material and Methods”.
- Instead of general bar graphs, the data should be presented by dot plots to show the position/distribution of individual biological replicates.
Answer: Thank you for your comment. We replaced the bar graphs used in Figure 1B, 1D, 1H, 2B, 2D, 2H, 3B, 3D, 4B, 4D, and 4E with dot plots graphs.
- The whole electrophoretogram/ blot images should be provided in the supplementary material to judge the results.
Answer: We had showed the raw data as Supple. Figure 1 of Supplementary Material in original version manuscript. We also attached Supple. Figure 6 of Supplementary Material in revised version as reviewer requested.
- On a separate note: the EV investigations should follow MISEV2018 guidelines published by International Society for Extracellular Vesicles (ISEV), (PMID: 30637094). It is not justified why authors call their isolated fractions exosomes but not the extracellular vesicles (EVs)?. To date there is not a single method available which could isolate exosomes alone. The isolated fraction is always a heterogenous population of EVs, no matter which method was used, and which material was used. Also, the markers such as CD63, CD9, CD81 and HSP70 etc are common for both exosomes and ectosomes (microvesicles). Similarly, the diameter or size detected by size detection instruments, cannot distinguish exosomes from microvesicles of same size. some microvesicles also show smaller diameters, like exosomes (thus EVs including small microvesicles and exosomes are called small EVs). If authors want to claim that the isolated fractions are exosomes then according to MISEV2018 guidelines (PMID: 30637094), at least two endosomal markers e.g. ALIX and TSG101 etc., should be identified along with surface markers (CD63, CD9,)., in order to verify that the vesicles are of endosomal origin i.e., exosomes. Otherwise, apply the term extracellular vesicles instead of exosomes.
Answer: We really appreciate your comment. We sincerely followed the MISEV2018 guidelines to isolate the extracellular vesicle such as exosome, and microvesicle. To address the issue, we added the molecular markers such as Alix, and TSC101 in Supple. Figures 1-4 as reviewer’s comment.
Others:
Although not of major concern, but import for readers, there are enormous errors in the text. Few examples are provided below but are not limited to;
- lack of nutrient, glucose, tuberous sclerosis complex (TSC) regulates – revise the phrase. Should it be…. tuberous sclerosis complex (TSC) is regulated by lack of nutrient, such as glucose via negative regulation of………….?
Answer: Thank you for your comment. To avoid confusion, the above sentence has been modified to the following sentence.
Tuberous sclerosis complex (TSC) regulates the cell viability via the negative regulation of the mammalian target of rapamycin complex (mTORC1) during glucose deprivation.
- two different type inhibitors – two different types of inhibitors
Answer: Thank you for your kind comment. We corrected “two different type inhibitors” to “two different types of inhibitors” in abstract.
iii. page 2, line 61: targeting mTORC1 becomes a the most available….
Answer: We appreciate your kind comments. We corrected the part into the following sentence.
“Therefore, targeting mTORC1 is one of the best ways to suppress the TSC function.”
- method of Richard J → Richard J et al.,
Answer: Thank you for your kind comment. We corrected “Richard J” to “Richard J. et al.” in Material and Methods.

Reviewer 3 Report
In this paper, authors indicated that mTORC1 inhibition can increased exome biogenesis, exosome secretion, and cell viability in TSC-deficient cells upon glucose deprivation. However, I think current data provided in this paper can not support their ideas sufficiently.
Figure 1-2, author should use one TSC-competent cell lines as control.
Figure 1E&F, 2E&F, author should add "Glc(-) DMSO"data in these figure.
Figure 3, author should use control siRNA treatment group as control.
Figure 4F, author should provide WB figure of total caspase-3.
Author Response
Dear Dr. Editor
Thank you for giving us the opportunity to revise our manuscript entitled “Exosome Release by Glucose Deprivation Is Important for the Cell Viability of TSC-null Cells” (Manuscript Number: Cells-1853422). We appreciate the reviewer’s constructive and insightful comments. We have revised the manuscript accordingly, and would like to re-submit it for your consideration. We have addressed the comments raised by the reviewers, and the amendments are highlighted in red in the revised manuscript. We hope that the revised version of the manuscript is now acceptable for publication in Cells journal. Point-by-point responses to the reviewers’ comments are listed below this letter.
Reviewer 3
In this paper, authors indicated that mTORC1 inhibition can increased exome biogenesis, exosome secretion, and cell viability in TSC-deficient cells upon glucose deprivation. However, I think current data provided in this paper can not support their ideas sufficiently.
- Figure 1-2, author should use one TSC-competent cell lines as control.
Answer: Answer: Thank you for your kind comment. We performed the corresponding experiment to the reviewer’s request using MEF-WT cells under the same condition. The levels of CD63 and CD81 used in original version and the levels of Alix and TSG101 added in revised version were unaffected under the same condition in MEF-WT cells when compared to that of TSC2-/- and TSC1-/- cells. Exosome release was also unaffected under the same condition in MEF-WT cells. We described the above in result section and added the explanation in Supple. Figure 2. (Please see the paragraph marked in red on page 5, line 12-16)
- Figure 1E&F, 2E&F, author should add "Glc(-) DMSO"data in these figure.
Answer: We had described why we couldn’t detect the TEM images and nanosize particle measured by NanoSight NS3000 from Malvern in the result section of original version. We would add the honorable data as reviewer requested if it were obtained. However, we failed to isolate extracellular vesicle (EV) particle in DMSO treatment under glucose deprivation condition. We speculate that the reason may be few extracellular vesicles in this condition.
- Figure 3, author should use control siRNA treatment group as control.
Answer: Thank you for your kind comment. We performed the corresponding experiment using HEK293 cells transfected with scrambled siRNAs under the same conditions. The levels of CD63 and CD81 as well as levels of Alix and TSG101 were unaffected under the same condition in HEK293 cells transfected with scrambled siRNAs when compared to that of HEK293 transfected cells with either TSC2 or TSC1 siRNAs. Exosome release was not changed in MEF-WT cells under the same condition. We described the mention above in result section and added the explanation in Supple. Figure 5. (Please see the paragraph marked in red on page 9, line 7 from bottom)
- Figure 4F, author should provide WB figure of total caspase-3.
Answer: Thank you for your kind comment. We inserted the levels of total caspase 3 and cleaved caspase 3 in panel F of Figure 4. The raw data was showed in Supple. Figure 6 in Supplemental Figure.

Round 2
Reviewer 1 Report
Thank you for the revisions made to the manuscript. I would just advise to edit more carefully. There are a few time you have uL instead of the greek symbol insert for microliter.
Reviewer 2 Report
In the revised version, the authors have performed the suggested experiments including the control cell line, and have supplemented the manuscript with additional results.
While the manuscript is substantially improved, it still remains a misunderstanding regarding the stress factors. The exosomes field has reportedly established that cell culture stress conditions bythemselves effect the exosome release, cell viability e.g, hypoxia, glucose deprivation, oxidative stress etc. thus by adding additional inhibitors, it is important to perform control experiments to rule out the chances whether the effects are merely due to the inhibitor, or stress or both, or one is dependent on other.
However, in the revised manuscript the authors have explained well the rationale of the study.
On a separate note, the authors showed exosome markers, but it does not rule out the presence of other populations of EVs, CD63 is shared between microvesicles and exosomes. and small sizes are also shared by heterogenous population of EVs. Hence the purpose of using a general term extracellular vesicles instead of exosomes as per MISEV2018 guidelines.
At line 217, please consider revising the following phrase ''treatment under in glucose''.
Reviewer 3 Report
Authors addressed my concerns properly, I have no more suggestion.